# Developing a Comprehensive Shipment Policy through Modified EPQ Model Considering Process Imperfections, Transportation Cost, and Backorders

**Waseem Sajjad** [1], **Misbah Ullah** [1,*], **Razaullah Khan** [2] and **Mubashir Hayat** [3,*]

1 Department of Industrial Engineering, University of Engineering and Technology, Peshawar 25120, Pakistan; waseemsajjad.ces@kpogcl.com.pk
2 Department of Engineering Management, University of Engineering and Applied Sciences, Swat 19060, Pakistan; razaullah@suet.edu.pk
3 Department of Mechanical Engineering, Campus Jalozai, University of Engineering and Technology, Peshawar 24240, Pakistan
* Correspondence: misbah@uetpeshawar.edu.pk (M.U.); mubashirhayat@uetpeshawar.edu.pk (M.H.)

**Abstract:** *Background:* Determining the optimum shipment quantity in a traditional production system is a competitive business dimension, and developing a reliable shipment policy is decisive for long-term objectives. Currently, significant research in this domain has mainly focused on the optimum shipment lot sizing in a perfect production system without considering the imperfections in the production processes and logistics. It has been well established that the real production inventory system acts as an imperfection in the overall production management loop. *Methods:* This research deals with designing a new shipment policy considering the imperfections in the production processes and undertaking some influential factors, such as the transportation cost, the actual production inventory, defective items, and backorders. *Results:* In the developed mathematical framework, the lot-sizing problems, imperfections in the production processes, retailers, and distributors are considered with equal-sized shipment policy to attain pragmatic and real-time results. *Conclusions*: The developed framework considers an all-unit-discount transportation cost structure. The numerical computations, as well as sensitivity analysis, are performed to point out the specifications and validation of the proposed model.

**Keywords:** economic production quantity; imperfection in the process; production management; backorders; transportations costs; supply chain management





## 1. Introduction

In the present socio-economic scenario, there is a consistent tug-of-war between the optimum order quantity and the shipment cost of items produced in any production setup. Considering the changing customers' requirements, production firms are constantly updating their shipment policies to optimize economic order quantities to enhance net profits. In this scenario, production shipment policy acts as an imperative parameter for any firm to survive in a high-paced competitive environment. This research focuses on a comprehensive shipment policy, considering the most significant parameters, including backorders in a shipment, imperfections in operations, and a reworked item in a production system. To minimize costs, different researchers have modeled several optimal batch problems considering different production conditions. For instance, Harris [1] is among the pioneers who developed the Economic Orders Quantity (EOQ) inventory model. The second most important model was developed by Taft [2]. This model is called the Economic Production Quantity (EPQ) model. Subsequently, these models were modified and expanded by other researchers [3–6]. The research has shown that small perturbations in the parameters of the EOQ and EPQ models do not impose any significant impact on the solution of a

problem [7–9]. Owing to this, the Economic Production Quantity (EPQ) model emerged as an optimal substitute that shows promising results for a production environment when applied with some assumptions.

In an actual production environment, the system runs with some imperfections. The imperfections in a production system produce low-quality items for several reasons—namely, defects in raw materials, changes in machine capabilities, backorders, rework, and differences in the experience of the operators. Some studies are available in the literature in which the proposed models have considered these imperfections. For example, Jamal et al. [3] studied the EPQ model to obtain the optimum batch size. The proposed model considers a re-work process after several production cycles. Expanding the contributions of Jamal et al. [3], Sarkar et al. [4] formulated the same problem with additional terms of backorders. The model proposed by Cardenas-Barron et al. [7] encompasses numerous parameters. The model addresses the reworked production quantities and other production system defects. Chang et al. [8] proposed a mathematical model for total time consumption in a production system involving manual and automatic operations. Wee et al. [5] adopted the same methodology and developed a model which considered the development of refurbished products with non-conformities. It was concluded that in repeated manufacturing cycles, there is an effective way to reprocess faulty and defective products. The data obtained confirmed that the critical aspects could be more related to the manufacturing cost and the service expenditures of the process.

An identical model was presented by Sarkar et al. [10], which focused on the inflation effect. It was shown that the prolonged use of the manufacturing units could potentially damage the smooth operating of the system, i.e., it could produce defects in the system. The focus of the research was on how to overcome the defects produced during smooth operations and to reprocess the defective products. The demands of overtime on the workers could be a cause of defects in the system, or it could be other unknown reasons. The focus is on the prolonged reproducibility of the system using the reliability of the system. The decision variable of system reliability was used to hypothesize a new model consisting of the integrated entities that were organized to maintain a smooth operating system as well as to instantly remove defects. It was found that the model reduced the operating system's total costs. By means of an algebraic model, it was shown that three parallel distribution systems can be operated to attain considerably lower production costs.

Swenseth et al. [11] derived a mathematical model for reducing overall production costs considering a large shipment volume. Ertogral et al. [9] established a model to deliver products to retailers considering the unit transportation cost. Ghasemy Yaghin et al. [12] developed a comprehensive model for an optimum number of lot sizes and quantities of various values. Ekici et al. [13] assumed certain limitations on the production batch size and proposed a model for varying customer demands under several manufacturing settings. Geri et al. [14] developed a model for stock policy in which the single manufacturer shipped single types of product to the retailer in a fixed lot size. Tseng et al. [15] reported that transportation cost is the key to economic growth, and it represents 6.5 percent of the market revenues. Rodigue et al. [16] assessed the conditions driving the global forms of production, distribution, and transport mainly by looking at the levels of geographical and functional integration of global production networks given the high level of disintegration within them.

Cardenas-Barrón [17] presented a simple derivation of the work presented by Jamal et al. [3]. Differential calculus was used to find the optimal solution to the problem, and the derivation was based on algebraic derivation. The results obtained were equivalent to the results obtained by Jamal et al. [3]. Cárdenas-Barrón et al. [18] extended the work of Chiu et al. [19,20] by determining both the optimal number of shipments and the optimal replenishment lot size. The solution of Cárdenas-Barrón et al. [18] presented better results than Chiu et al. [19,20]. Kun-Jen Chung [21] studied the work of Cárdenas-Barrón et al. [7] and presented a solution procedure for finding the optimal solution of the total cost. Goyal et al. [22] considered the problem of determining the economic production and shipment policy of a product

supplied by a vendor to a single buyer to minimize the total costs incurred by the vendor and the buyer. Sana et al. [23] developed a framework of production policy to find the optimal safety stock, production lot size, and optimal production rate. Sarkar et al. [6] derived mathematical models to obtain the optimal cycle time to minimize the annual total relevant cost.

To reduce the production of imperfect products, the production systems must be highly reliable to cope with the changing shipment demands and supply. Some substantial studies have considered production and system reliability parameters. For instance, Sarkar et al. [24] considered the production cost, development cost, and material cost as dependent reliability parameters. Sarkar et al. [25] developed a production-inventory model for item deterioration in a two-echelon supply chain management. The objectives of the study were to obtain minimum cost and optimum lot size for three different models with an integer number of deliveries. Sarkar et al. [26] presented an economic manufacturing quantity model for stock-dependent demand in an imperfect production process in which unit production cost was employed as a function of production rate and reliability parameters. Extending their work, a new model presented by Sarkar et al. [27] adopted considered preventive and corrective maintenance, and safety stock for repair times.

Likewise, Sarkar et al. [28] also developed models for optimum batch quantity in a multi-stage system with a rework process for two operational policies. The first policy deals with rework within the same cycle with no shortage, and the second policy deals with the rework done after several cycles incurring shortages in each cycle. Taleizadeh et al. [29] developed an economic production and inventory model in a three-layer supply chain for a single-product and general demand functions. Hayat et al. [30] developed a shipment policy for an imperfect production setup with transportation costs taken into consideration. The model analyzed lot-sizing for manufacturers and retailers with imperfections in terms of equal-sized shipments.

Wanzhu et al. [31] worked on optimum production lot sizing and implemented the strategy in a quick response manufacturing setup. The statistical evaluations induced optimized production scheduling and significant improvement in lead time, including shipment time and costs. Abdul et al. [32] developed a transportation cost model to analyze the supply chain efficiency, including the shipment of products with different lot sizes and in-process inventory. They proposed a mathematical model based on metaheuristics, and the result was supportive in a closed-loop supply chain context. Asad et al. [33] also highlighted some important parameters which are imperative for the components of supply chain integration considering shipment of products to retailers. Waqas et al. [34] studied the prospective dimensions of production flexibility at their interface with the integrated functional units. The work showed that optimal production flexibility is important for developing a reliable shipment policy. With system flexibility, the implementation of an integrated advanced manufacturing planning and execution system, which could support shorter product cycles with real-time monitoring of shipment processes, is imperative [35].

Guchhait et al. [36] developed and optimized the cost model by addressing the defective products, backorders, and warranty policy. Taleizadeh et al. [37] developed a stochastic inventory control model with partial backordering and introduced supply disruption. Snyder et al. [38] proposed a model that considers the variance of the variable lead time-dependent demand. They estimated mean and variance through smoothing methods. Hsiao et al. [39] also employed a stochastic demand and developed a model for a single vendors that considers a delay in transportation. Dominguez et al. [40] presented a model with variable lead time that addresses the dynamic property of the closed-loop supply chain system and the multi-echelon supply chain. Haeussler et al. [41] also introduced a model by employing optimization techniques. Malik and Sarkar [42] introduced a game strategy and stochastic lead time demand to reduce the total expected cost. The contributions of researchers in similar areas are presented in Table 1.

**Table 1.** Comparison of the proposed research with existing literature.

| Reference | IP [a] | B [b] | SP [c] | SIP [d] | SIPD [e] |
|---|---|---|---|---|---|
| Goyal et al. [22] | ✗ | ✗ | ✓ | ✗ | ✗ |
| Swenseth et al. [11] | ✗ | ✗ | ✓ | ✗ | ✗ |
| Jamal et al. [3] | ✓ | ✗ | ✗ | ✗ | ✗ |
| Ertogral et al. [9] | ✗ | ✗ | ✓ | ✗ | ✗ |
| Cárdenas-Barrón [17] | ✓ | ✗ | ✗ | ✗ | ✗ |
| Cárdenas-Barrón et al. [7] | ✓ | ✓ | ✗ | ✗ | ✗ |
| Sana et al. [23] | ✓ | ✗ | ✗ | ✗ | ✗ |
| Chung [21] | ✓ | ✓ | ✗ | ✗ | ✗ |
| Chang et al. [8] | ✓ | ✓ | ✗ | ✗ | ✗ |
| Cárdenas-Barrón et al. [18] | ✓ | ✗ | ✗ | ✗ | ✗ |
| Sarkar et al. [4] | ✓ | ✓ | ✗ | ✗ | ✗ |
| Ghasemy Yaghin et al. [12] | ✗ | ✗ | ✓ | ✗ | ✗ |
| Taleizadeh et al. [29] | ✗ | ✗ | ✓ | ✗ | ✗ |
| Ekici et al. [13] | ✗ | ✗ | ✓ | ✗ | ✗ |
| Geri et al. [14] | ✗ | ✗ | ✓ | ✗ | ✗ |
| Hayat et al. [30] | ✓ | ✓ | ✓ | ✓ | ✗ |
| Our proposed model | ✓ | ✓ | ✓ | ✓ | ✓ |

[a] Imperfection in process, [b] Backordering, [c] Shipment policy, [d] Shipment with imperfection in process, [e] Shipment with imperfection in the process considering distributors.

The proposed work expands the most recent study of Hayat et al. [30] by developing a shipment policy for the defective manufacturing system. None of the previous studies mentioned in Table 1 have considered the distributor in their model development for imperfect systems. However, in real-life cases, the distributor is as an important stakeholder and must not be ignored in the mathematical formulation. In this context, the proposed work extends the previous research with the inclusion of the distributor in the model development. For the proposed scenario, the inventory flow with rework and backorders is illustrated in Figure 1.

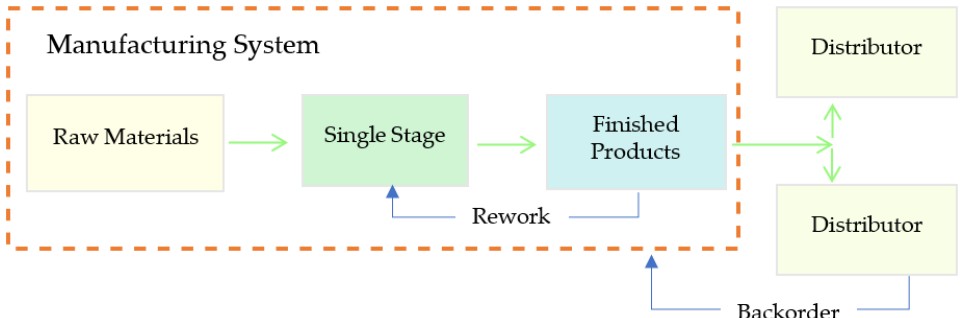

**Figure 1.** Inventory flow in an imperfect production setup.

The rest of this paper is organized as follows: Section 2 specifies the case for model development and provides the parameters and assumptions to be taken in the model development. Mathematical models for two of the proposed cases are developed in Section 3. For the purpose of having more insight into the developed models, Section 4 presents numerical computation and sensitivity analysis. Finally, this work is summarized and future directions are provided in Section 5.

## 2. Model Specifications

In the proposed model, the designed framework considers a production system with a single manufacturer, distributor, and retailer. The manufacturer makes finished products by processing raw materials in a single-stage production setup. This is explained in Figure 2. In Figure 2, the mathematical expressions for associated costs, except distributor ordering cost, are based on the formulas derived by Sarkar et al. [4], Cardenas-Barron et al. [7], and

Hayat et al. [30], while the expressions for optimum size and number of shipments are the output of our developed models. In a manufacturing setup, imperfect products are re-processed in the same production cycle to make acceptable quality products. In this model, the finished products are delivered to the distributor and then to the retailer. The proposed model also allows backorders, and the manufacturer, distributor, and retailer are accountable for transportation costs. The shipment of finished products to the distributors and delivery to retailers occurs in a perfect shipment system.

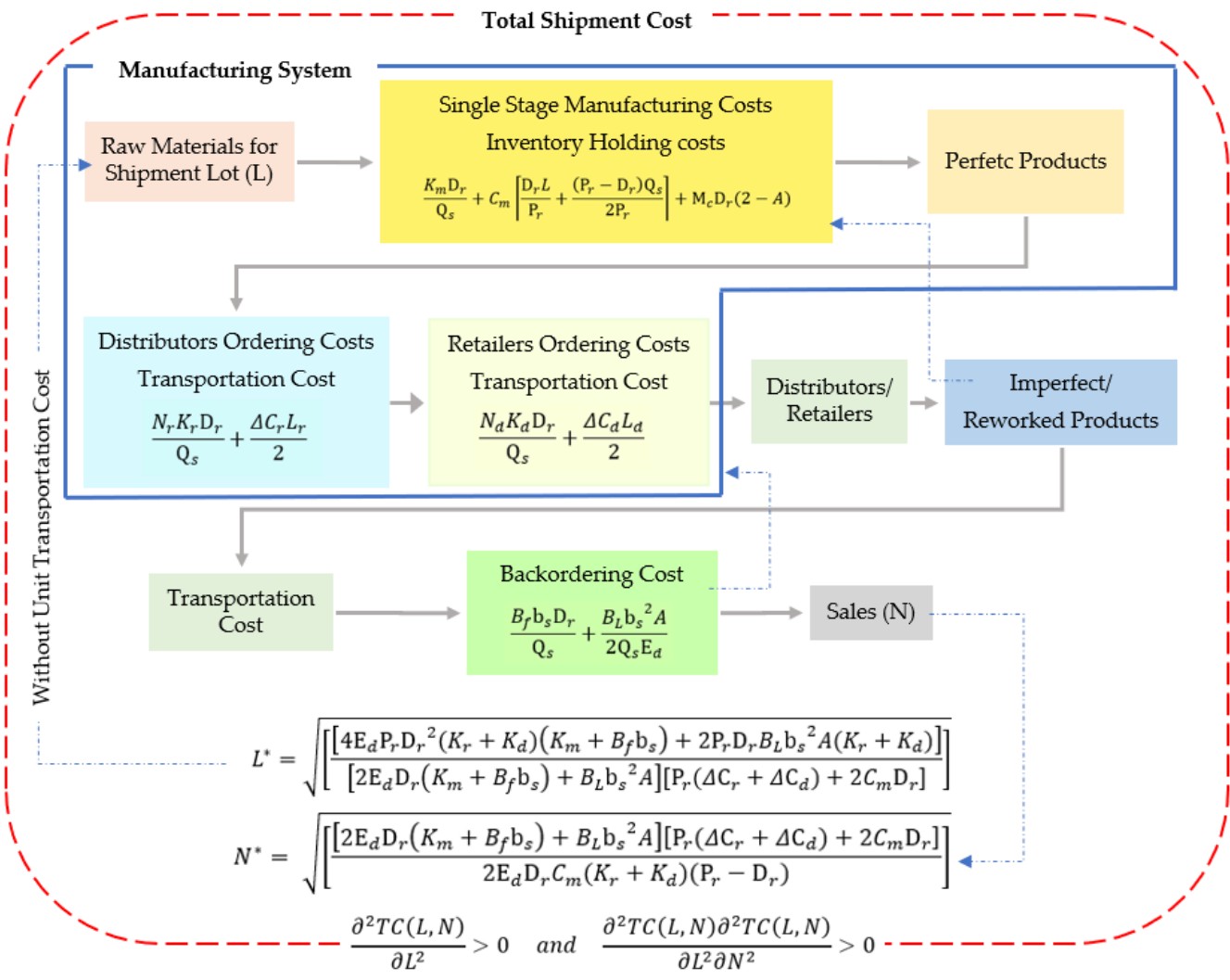

**Figure 2.** Modified EOQ model for optimized shipment in an imperfect production setup with transportation to distributor and retailer.

The objectives of the proposed model are to:

- Consider manufacturer, distributor, and retailer for the development of the economic production quantity model with imperfections in the process.
- Develop the shipment policy for the proposed model.
- Evaluate the effect of the transportation cost on the overall cost of the system.
- The mathematical model is characterized by the following parameters and variables:

### 2.1. Parameters

- $P_r$ Production rates
- $D_r$ Demand rate
- $Q_s$ Batch size
- $K_m$ Productions setups cost

- $M_c$ Manufacturing costs of item
- $K_d$ Ordering cost of the distributor
- $b_s$ Backorders size
- $K_r$ Ordering costs of the retailer
- C Inventory carry cost per product per unit of time
- $C_s$ Inventory holding cost of the system
- $B_L$ Linear backorder costs per item per unit of time
- $C_r$ Retailers inventory carry cost
- $T_b$ Back-ordering transportations cost
- $C_d$ Inventory carry costs of the distributor
- $B_f$ Fixed back-ordering costs per product
- $T_u$ Transportation cost per unit
- L Shipment lot size
- N Number of shipments
- $E_d$ Anticipated rate of ratio defectives product in each cycle
- TC Totals costs per unit of times

*2.2. Assumptions*

- Productions and demand would be constants and identified in the planning limit.
- Production rates would be larger than demand rates.
- The ratio of the imperfect product is a random variable in each production cycle. It would follow three various distribution functions such as uniform, beta, and triangle, modelled by static transportation cost.
- The manufacturers screen 100% of the product after each production cycle to produce good quality products and the screening/inspection cost of the products are ignored.
- With each cycle, there would be no waste product and all imperfect products are supposed to be reworked again to produce a good quality product.
- Holding/carrying costs are taken based on average inventory.
- The model considers two kinds of back-order costs. The first is a linear back-ordering cost which is applied with the average and fixed backorder. The second back-ordering cost is applied with the extreme back-order levels.
- For each manufacturing setup, there are fixed transportation and backorders for the first model.
- The reworked and production process remains unchanged if the production rate remains unchanged.
- The inventory storage place and accessibility of assets are not constrained at any level.
- The model is applicable for a single type of item.

## 3. Model Formulation

As the model considers the shipment of finished products from the manufacturer to distributor and then to the retailer, the model is composed of different types of costs, i.e., manufacturer setup cost, distributor ordering cost, retailer ordering cost, inventory carrying cost of the manufacturer, distributor, and retailer, fixed and linear back-ordering costs, production cost, and unit transportation cost. In this setup, imperfect items are also produced and are reprocessed in the same production cycle into perfect items. It is assumed that this defect rate follows uniform distribution and has the same production cost whenever defective products are reprocessed.

*3.1. Case 1: Shipment Policy for an EPQ Model Considering Backorders without Unit Transportation Cost*

This case considers a scenario in which the unit transportation cost is either as considered insignificant or is included in the ordering cost. Therefore, in this case, the transportation cost would not be considered as an independent function in the development of shipment policy. Hence, for this case, the total cost function is as follows:

Total Cost = Manufacturer setups costs + Retailer orderings costs + Distributor orderings costs + Manufacturer inventory holding costs + Inventory carrying cost of retailer + Inventory holding/carrying cost of distributor + Fixed back-ordering cost + Linear back-ordering costs+ Production cost.

Incorporating the values of these cost functions, the total cost is equal to:

$$TC(L,N) = \frac{K_m D_r}{Q_s} + \frac{N_r K_r D_r}{Q_s} + \frac{N_d K_d D_r}{Q_s} + C_m \left[ \frac{D_r L}{P_r} + \frac{(P_r - D_r)Q_s}{2P_r} \right] + \frac{\Delta C_r L_r}{2}$$
$$+ \frac{\Delta C_d L_d}{2} + \frac{B_f b_s D_r}{Q_s} + \frac{B_L b_s{}^2 A}{2Q_s E_d} + M_c D_r (2 - A) \tag{1}$$

As batch size ($Q_s$) is comprised of shipment lot size (*L*) and number of shipments (*N*), replacing it and rearranging Equation (1) results in:

$$TC(L,N) = \frac{(K_m + N(K_r + K_d))D_r}{NL} + C_m \left[ \frac{D_r L}{P_r} + \frac{(P_r - D_r)NL}{2P_r} \right] + \frac{\Delta C_r L_r}{2} + \frac{\Delta C_d L_d}{2}$$
$$+ \frac{B_f b_s D_r}{NL} + \frac{B_L b_s{}^2 A}{2NLE_d} + M_c D_r (2 - A) \tag{2}$$

The system equation cost includes the two variables, i.e., *L* and *N*. Therefore, to reduce the total costs, the second order Hessian's matrix would be a positive definite, which implies that all the principle minors are positive. Therefore, the sufficient conditions for this problem are:

$$\frac{\partial^2 TC(L,N)}{\partial L^2} > 0 \quad and \quad \frac{\partial^2 TC(L,N) \partial^2 TC(L,N)}{\partial L^2 \partial N^2} > 0$$

Taking the partial derivative of Equation (2) with respect to *L* and *N*:

$$\frac{\partial TC}{\partial L} = -\frac{K_m D_r}{NL^2} - \frac{K_r D_r}{L^2} - \frac{K_d D_r}{L^2} + \frac{C_m D_r}{P_r} + \frac{C_m (P_r - D_r)N}{2P_r} + \frac{\Delta C_r}{2} + \frac{\Delta C_d}{2} - \frac{B_f b_s D_r}{NL^2} - \frac{B_L b_s{}^2 A}{2NL^2 E_d} \tag{3}$$

And

$$\frac{\partial TC}{\partial N} = -\frac{K_m D_r}{N^2 L} + \frac{C_m (P_r - D_r)L}{2P_r} - \frac{B_f b_s D_r}{N^2 L} - \frac{B_L b_s{}^2 A}{2N^2 L E_d} \tag{4}$$

From sufficient conditions, Equations (3) and (4) are transformed into:

$$\frac{\partial^2 TC}{\partial L^2} = \left( \frac{2}{NL^3} \right) \left[ \left( K_m + B_f b_s \right) D_r + \frac{B_L b_s{}^2 A}{2E_d} \right] + \frac{2K_r D_r}{L^3} + \frac{2K_d D_r}{L^3} \tag{5}$$

$$\frac{\partial^2 TC}{\partial N^2} = \left( \frac{2}{N^3 L} \right) \left[ \left( K_m + B_f b_s \right) D_r + \frac{B_L b_s{}^2 A}{2E_d} \right] \tag{6}$$

$$\frac{\partial^2 TC}{\partial L \partial N} = \frac{K_m}{N^2 L^2} + \frac{C_m (P_r - D_r)}{2P_r} + \frac{B_f b_s D_r}{N^2 L^2} + \frac{B_L b_s{}^2 A}{2N^2 L^2 E_d} \frac{\partial^2 TC}{\partial L \partial N}$$
$$= \left( \frac{1}{N^2 L^2} \right) \left[ \left( K_m + B_f b_s \right) D_r + \frac{B_L b_s{}^2 A}{2E_d} \right] + \frac{C_m (P_r - D_r)}{2P_r} \tag{7}$$

$$\left( \frac{\partial^2 TC}{\partial L \partial N} \right)^2 = \left( \frac{1}{N^2 L^2} \right)^2 \left[ \left( K_m + B_f b_s \right) D_r + \frac{B_L b_s{}^2 A}{2E_d} \right]^2 + \left[ \frac{C_m (P_r - D_r)}{2P_r} \right]^2$$
$$+ 2 \left( \frac{1}{N^2 L^2} \right) \left[ \left( K_m + B_f b_s \right) D_r + \frac{B_L b_s{}^2 A}{2E_d} \right] \left[ \frac{C_m (P_r - D_r)}{2P_r} \right] \tag{8}$$

$$\frac{\partial^2 TC \partial^2 TC}{\partial L^2 \partial N^2} = \left( \frac{4}{N^4 L^4} \right) \left[ \left( K_m + B_f b_s \right) D_r + \frac{B_L b_s{}^2 A}{2E_d} \right]^2 + \left( \frac{4K_r D_r}{N^3 L^4} + \frac{4K_d D_r}{N^3 L^4} \right) \left[ \left( K_m + B_f b_s \right) D_r + \frac{B_L b_s{}^2 A}{2E_d} \right] \tag{9}$$

Therefore,

$$\frac{\partial^2 TC \partial^2 TC}{\partial L^2 \partial N^2} - \left(\frac{\partial^2 TC}{\partial L \partial N}\right)^2$$

$$= \left(\frac{4}{N^4 L^4}\right) \left[\left(K_m + B_f\, b_s\right) D_r + \frac{B_L\, b_s^2 A}{2E_d}\right]^2 + \left(\frac{4K_r D_r}{N^3 L^4} + \frac{4K_d D_r}{N^3 L^4}\right) \left[\left(K_m + B_f b_s\right) D_r + \frac{B_L b_s^2 A}{2E_d}\right]$$

$$- \left(\frac{1}{N^2 L^2}\right)^2 \left[\left(K_m + B_f\, b_s\right) D_r + \frac{B_L\, b_s^2 A}{2E_d}\right]^2 - \left[\frac{C_m (P_r - D_r)}{2P_r}\right]^2$$

$$- 2\left(\frac{1}{N^2 L^2}\right)\left[\left(K_m + B_f\, b_s\right) D_r + \frac{B_L\, b_s^2 A}{2E_d}\right]\left[\frac{C_m (P_r - D_r)}{2P_r}\right] \tag{10}$$

$$X = \left(\frac{1}{N^2 L^2}\right)\left[\left(K_m + B_f b_s\right) D_r + \left[\frac{B_L b_s^2 A}{2E_d}\right]\right],$$
$$Y = \left(\frac{4K_r D_r}{N^3 L^4} + \frac{4K_d D_r}{N^3 L^4}\right)\left[\left(K_m + B_f b_s\right) D_r + \frac{B_L b_s^2 A}{2E_d}\right],\ Z = \left[\frac{C_m (P_r - D_r)}{2P_r}\right] \tag{11}$$

Hence,

$$\frac{\partial^2 TC \partial^2 TC}{\partial L^2 \partial N^2} - \left(\frac{\partial^2 TC}{\partial L \partial N}\right)^2 = 4X^2 + Y - X^2 - Z^2 - 2XZ \tag{12}$$

$$\frac{\partial^2 TC \partial^2 TC}{\partial L^2 \partial N^2} - \left(\frac{\partial^2 TC}{\partial L \partial N}\right)^2 = 3X^2 + Y - Z^2 - 2XZ \tag{13}$$

To simplify the above equation, the optimality conditions are satisfied if the sufficient expression $3X^2 + Y - Z^2 - 2XZ$ is greater than 0. The first partial derivatives with respect to variables $N$ and $L$ are distinctly equal to zero, so that the optimum points would be obtained. Therefore, putting Equation (2) equal to Zero and simplifying for $L^*$ results in:

$$0 = -\frac{(K_m + N(K_r + K_d)) D_r}{NL^2} + \frac{C_m D_r}{P_r} + \frac{C_m (P_r - D_r) N}{2P_r} + \frac{\Delta C_r}{2} + \frac{\Delta C_d}{2} - \frac{B_f b_s D_r}{NL^2} - \frac{B_L b_s^2 A}{2NL^2 E_d} \tag{14}$$

$$L^* = \sqrt{\frac{\left[2E_d D_r \left(K_m + N(K_r + K_d) + B_f b_s\right) + B_L b_s^2 A\right]}{\left[2NE_d \left(C_m \left(\frac{D_r}{P_r} + \frac{(P_r - D_r) N}{2P_r}\right) + \frac{\Delta C_r}{2} + \frac{\Delta C_d}{2}\right)\right]}} \tag{15}$$

Now, putting Equation (15) equivalent to 0 and simplifying for $N^*$ results in:

$$0 = -\frac{K_m D_r}{N^2 L} + \frac{C_m (P_r - D_r) L}{2P_r} - \frac{B_f b_s D_r}{N^2 L} - \frac{B_L b_s^2 A}{2N^2 L E_d} \tag{16}$$

$$N^* = \frac{\sqrt{\left[8E_d^2 P_r D_r C_m \left(K_m + B_f b_s\right)(P_r - D_r) + 4E_d P_r B_f b_s^2 A C_m (P_r - D_r)\right]}}{2L E_d C_m (P_r - D_r)} \tag{17}$$

Solving Equations (15) and (17) simultaneously results in:

$$L = \frac{\sqrt{\left[8E_d^2 P_r D_r C_m \left(K_m + B_f b_s\right)(P_r - D_r) + 4E_d P_r B_L b_s^2 A C_m (P_r - D_r)\right]}}{2N E_d C_m (P_r - D_r)} \tag{18}$$

Now, subtracting Equation (18) from Equation (15):

$$0 = \sqrt{\left[\frac{\left[2E_d D_r \left(K_m + N(K_r + K_d) + B_f b_s\right) + B_L b_s^2 A\right]}{2N E_d \left[C_m \left(\frac{D_r}{P_r} + \frac{(P_r - D_r) N}{2P_r} + \frac{\Delta C_r}{2} + \frac{\Delta C_d}{2}\right)\right]}\right]}$$
$$- \frac{\sqrt{\left[8E_d^2 P_r D_r C_m \left(K_m + B_f b_s\right)(P_r - D_r) + 4E_d P_r B_L b_s^2 A C_m (P_r - D_r)\right]}}{2N E_d C_m (P_r - D_r)} \tag{19}$$

$$(0)^2 = \left(\sqrt{\left[\left[\frac{\left[2E_d D_r \left(K_m + N(K_r + K_d) + B_f b_s\right) + B_L b_s^2 A\right]}{2N E_d \left[C_m \left(\frac{D_r}{P_r} + \frac{(P_r - D_r) N}{2P_r}\right) + \frac{\Delta C_r}{2} + \frac{\Delta C_d}{2}\right]}\right] - \left[\frac{4E_d\, P_r D_r \left(K_m + B_f b_s\right) + 2P_r B_L b_s^2 A}{2N^2 E_d C_m (P_r - D_r)}\right]\right]}\right)^2 \tag{20}$$

$$\frac{\left[4E_dP_rD_r\left(K_m + B_fb_s\right) + 2P_rB_Lb_s{}^2A\right]}{2N^2E_dC_m(P_r - D_r)} = \frac{\left[2E_dD_r\left(K_m + N(K_r + K_d) + B_fb_s\right) + B_Lb_s{}^2A\right]}{2NE_d\left[\frac{2C_mD_r + C_mN(P_r - D_r) + P_r(\Delta C_r + \Delta C_d)}{2P_r}\right]} \tag{21}$$

$$N^* = \sqrt{\left[\frac{\left[2E_dD_r\left(K_m + B_fb_s\right) + B_Lb_s{}^2A\right]\left[P_r(\Delta C_r + \Delta C_d) + 2C_mD_r\right]}{2E_dD_rC_m(K_r + K_d)(P_r - D_r)}\right]} \tag{22}$$

Putting the value of $N^*$ in Equation (18) and simplifying results into:

$$L = \frac{\sqrt{\left[8E_d{}^2P_rD_rC_m\left(K_m + B_fb_s\right)(P_r - D_r) + 4E_dP_rB_Lb_s{}^2AC_m(P_r - D_r)\right]}}{2E_dC_m(P_r - D_r)\sqrt{\left[\frac{\left[2E_dD_r\left(K_m + B_fb_s\right) + B_Lb_s{}^2A\right]\left[P_r(\Delta C_r + \Delta C_d) + 2C_mD_r\right]}{2E_dD_rC_m(K_r + K_d)(P_r - D_r)}\right]}} \tag{23}$$

$$L^* = \sqrt{\left[\frac{\left[4E_dP_rD_r{}^2(K_r + K_d)\left(K_m + B_fb_s\right) + 2P_rD_rB_Lb_s{}^2A(K_r + K_d)\right]}{\left[2E_dD_r\left(K_m + B_fb_s\right) + B_Lb_s{}^2A\right]\left[P_r(\Delta C_r + \Delta C_d) + 2C_mD_r\right]}\right]} \tag{24}$$

Equations (22) and (24) represent the optimum number of shipments and shipment lot size, respectively. The total cost obtained based on these optimum numbers will be the minimum total cost. Numeric computations and validations of these optimum points for the given values of the parameters have been carried out in detail in Section 4.

*3.2. Case 2: Shipment Policy for an EPQ Model Considering Back-Orders and Unit Transportation Cost*

In this case, we developed models for systems that consider the transportation costs as distinct cost functions instead of considering them as a portion of ordering costs. The transport costs are considered as a separate entity. These costs rely on item shipment size and all-unit discount structure. The transportation costs involved during shipment from manufacturer to distributor and formerly to the retailer are considered the unit transportation costs. In Table 2, the unit transportation cost structures are explained.

**Table 2.** Items unit transportation costs structure.

| Items Series | Transportation Cost per Item in Dollar's |
|:---:|:---:|
| $0 \le L < X_1$ | $T_{u0}$ |
| $X_1 \le L < X_2$ | $T_{u1}$ |
| $X_2 \le L < X_3$ | $T_{u2}$ |
| . . . | . . . |
| $X_{m-1} \le L < X_m$ | $T_{um-1}$ |
| $X_m \le L$ | $T_{um}$ |

Where $T_{u0} > T_{u1} > T_{u2} > \ldots > T_{um}$.

In Table 2, the cost structure specifies the range of lot sizes between 0 and $X_1$. Therefore, the transportation unit costs of producer to distributor and retailers are expressed as $T_{u0}$, $T_{u1}$, $T_{u2}$, and so on. Additionally, if the shipment lot size is greater than or equivalent to $X_m$, the unit transportation cost would be $T_{um}$. The lot size $Q \epsilon [X_i, X_{i+1}]$ transport costs of each item are equivalent. Consequently, for an assumed range, the equation for transportation costs is shown in Table 3.

**Table 3.** Equation for transportation cost.

| | | |
|---|---|---|
| $TC(L) =$ | $T_{u0}D_r$ | $L \in (0, \ X_1)$ |
| $TC(L) =$ | $T_{u1}D_r$ | $L \in (X_1, X_2)$ |
| $TC(L) =$ | $T_{u2}D_r$ | $L \in (X_2, \ X_3)$ |
| $\dots$ | | $\dots$ |
| $TC(L) =$ | $T_{um}D_r$ | $L \in (X_m, \ infinity)$ |

Hence, the total cost equation per unit time counting the transportations costs is:

$$C(L, N) = TC_1(L, N) + TC(L) \tag{25}$$

## 4. Numerical Computation and Sensitivity Analysis

To illustrate the specifications and provide additional insight into the model, this section explains the numerical computations and sensitivity analysis.

### 4.1. Numerical Example for Case 1

A manufacturer produces 550 items per year. The yearly demand placed by customers is 300 units. From the previous year's record, it is known that the manufacturer inventory carrying cost per product per unit of time is USD 50, retailer inventory carrying cost per product per unit of time is USD 5, distributor inventory carrying cost per product per unit of time is USD 5, linear backorder cost per product per unit of time is USD 10, and fixed backorder cost per product is USD 1. The manufacturing cost of a product is USD 7, the manufacturer production setup cost is USD 50, the retailer ordering cost is USD 5, and the backorder size is 33 items. If the values of other parameters are assumed as = 0.03 and b = 0.07 [4], then the optimum number of shipments, optimum quantity, and the total optimum cost can be calculated as follows:

$$A = 1 - \frac{a + b}{2} = 0.95$$

$$E_d = 1 - \frac{a + b}{2} - \frac{D_r}{P_r} = 0.4045$$

Putting the values in Equation (23) results in: $N^* = 5.97$. Therefore, $N^*$ can be either 5 or 6.

(1) For $N^* = 5$, the optimum lot size is $L^* = 10.87 \ units$, and the optimum total cost is $TC(L, N) = \$4115.65$.

(2) For $N^* = 6$ (3.19), the optimum lot size is $L^* = 9.61 \ units$, and the optimum total cost is $TC(L, N) = \$4112.14$.

Therefore, the optimal solution for $N^*$ and $L^*$ is:

$L^* = 9.61$ lot size,

$N^* = 6$ number of shipments and

$TC \ (L, N) = \$4112.14$ is the minimum total cost

Sensitivity analysis is important to establish the uncertainty of the model and evaluate the effect of the independent variables on the dependent variables under a given set of constraints and assumptions. In case 1, the results of sensitivity analyses of essential parameters are presented in Table 4 and explained in Figure 3a–k. Microsoft Excel has been used in model calculation as well as to perform the sensitivity analysis of the model. Figure 3a explains that the total system's cost decreases as the production rate decreases. It is reduced by up to 12.75% with a 50% decrease in the production rate. It is observed that the cost is most sensitive when the production rate is reduced below 25%. As shown in Figure 3b, the demand rate is also observed to be the most sensitive parameter among all other parameters. An increase in the demand value by 50% increases the system's cost by 40.41%, and a decrease in demand by 50% reduces the overall system cost by 31.47%.

This means that the total system cost is not affected significantly by the demand rate as compared to other parameters.

**Table 4.** Sensitivity analysis for basic parameters of the model presented in Case 1.

| Parameters | Variation | Variation (%) | Totals Costs | Variation in Total Cost (%) |
|---|---|---|---|---|
| $P_r$ | 687.5 | 25 | 4216.99 | 2.55 |
| | 825 | 50 | 4286.90 | 4.25 |
| | 275 | −50 | 3587.83 | −12.75 |
| | 412.5 | −25 | 3937.37 | −4.25 |
| $K_m$ | 25 | 25 | 3980.74 | 1.51 |
| | 37.5 | 50 | 4047.62 | 2.98 |
| | 62.5 | −50 | 4174.50 | −3.19 |
| | 75 | −25 | 4234.94 | −1.56 |
| $K_r$ | 6.25 | 25 | 4149.83 | 0.91 |
| | 7.5 | 50 | 4186.75 | 1.81 |
| | 2.5 | −50 | 4034.46 | −1.88 |
| | 3.75 | −25 | 4073.70 | −0.93 |
| $K_d$ | 6.25 | 25 | 4149.80 | 0.91 |
| | 7.5 | 50 | 4186.74 | 1.81 |
| | 2.5 | −50 | 4034.45 | −1.88 |
| | 3.75 | −25 | 4073.70 | −0.93 |
| $C_r$ | 6.25 | 25 | 4118.30 | 0.14 |
| | 7.5 | 50 | 4124.48 | 0.30 |
| | 2.5 | −50 | 4099.85 | −0.29 |
| | 3.75 | −25 | 4105.98 | −0.14 |
| $L_d$ | 6.0 | 25 | 4170.57 | 1.42 |
| | 7.2 | 50 | 4229.01 | 2.84 |
| | 2.4 | −50 | 3995.26 | −2.84 |
| | 3.6 | −25 | 4053.70 | −1.42 |
| $C_d$ | 6.25 | 25 | 4112.29 | 0.003 |
| | 7.5 | 50 | 4112.47 | 0.008 |
| | 2.5 | −50 | 4111.87 | −0.006 |
| | 3.75 | −25 | 4111.99 | −0.003 |
| $C_m$ | 62.5 | −50 | 4334.16 | −13.38 |
| | 75 | −25 | 4534.94 | 6.12 |
| | 25 | 25 | 3561.80 | 5.39 |
| | 37.5 | 50 | 3860.24 | 10.28 |
| $M_c$ | 8.75 | 25 | 4663.38 | 13.40 |
| | 10.5 | 50 | 5214.63 | 26.81 |
| | 3.5 | −50 | 3009.63 | −26.81 |
| | 5.25 | −25 | 3560.88 | −13.40 |

**Table 4.** *Cont.*

| Parameters | Variation | Variation (%) | Totals Costs | Variation in Total Cost (%) |
|---|---|---|---|---|
| a | 0.75 | 25 | 4119.15 | 0.17 |
| | 0.045 | 50 | 4126.17 | 0.34 |
| | 0.015 | −50 | 4098.09 | −0.34 |
| | 0.05 | −25 | 4105.11 | −0.17 |
| b | 0.085 | 25 | 4128.51 | 0.39 |
| | 0.105 | 50 | 4144.90 | 0.79 |
| | 0.045 | −50 | 4079.36 | −0.79 |
| | 0.0525 | −25 | 4095.75 | −0.39 |
| $D_r$ | 375 | 25 | 4855.11 | 17.38 |
| | 450 | 50 | 5807.37 | 40.41 |
| | 150 | −50 | 2834.42 | −31.47 |
| | 225 | −25 | 3474.02 | −16.01 |
| $B_L$ | 12.5 | 25 | 4167.57 | 1.34 |
| | 15 | 50 | 4223.00 | 2.69 |
| | 5 | −50 | 4001.27 | −2.69 |
| | 7.5 | −25 | 4056.70 | −1.34 |
| $B_f$ | 1.25 | 25 | 4155.05 | 1.04 |
| | 1.5 | 50 | 4197.96 | 2.08 |
| | 0.5 | −50 | 4026.30 | −2.08 |
| | 0.75 | −25 | 4069.22 | −1.04 |
| $L_r$ | 6.00 | 25 | 4170.57 | 1.42 |
| | 7.20 | 50 | 4229.01 | 2.84 |
| | 2.40 | −50 | 3995.26 | −2.84 |
| | 3.60 | −25 | 4053.70 | −1.42 |
| N | 7.5 | 25 | 4169.23 | 0.42 |
| | 9 | 50 | 4245.89 | 1.14 |
| | 3 | −50 | 4462.12 | −1.77 |
| | 4.5 | −25 | 4190.20 | −0.16 |
| L | 12.013 | 25 | 4188.74 | 1.86 |
| | 14. 41183 | 50 | 4318.27 | 5.01 |
| | 4.8051 | −50 | 4435.28 | −7.85 |
| | 7.307 | −25 | 4141.39 | −0.71 |

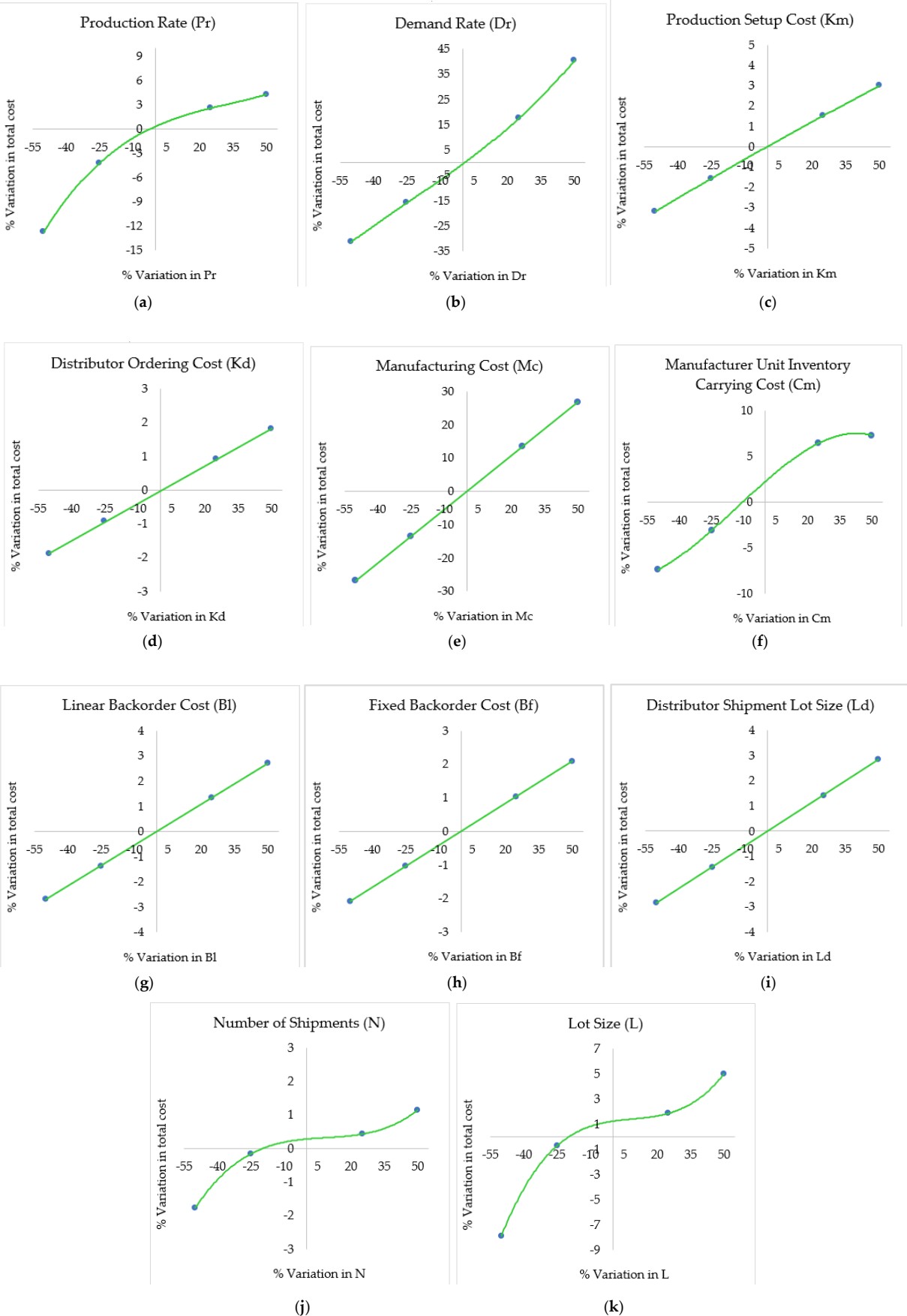

**Figure 3.** (**a**–**k** from top left to bottom right) Sensitivity analyses of important parameters (Case 1).

Figure 3c shows the affiliation of the production setup cost with the total system's cost. From the numerical figures, this parameter seems to be non-sensitive as the % variation in the total cost remains around 1.5% when increasing or decreasing the % of the setup cost value by 50%. Figure 3d illustrates the distributor ordering cost and retailer ordering cost. From the obtained figures, it is evident that these parameters are also very sensitive to the overall system's cost. In both types of ordering costs, the variation in system cost is reduced by about 1.8% only by increasing or decreasing the value of ordering cost either by 25% or 50%. As shown in Figure 3e, there is a linear relationship between the manufacturing production setup cost and the total cost. This parameter shows that the total system cost, which is the most sensitive, is reduced up to 26% when the machining cost is reduced up to 50%. As shown in Figure 3f, the manufacturer inventory carrying costs also affect the total system's cost appreciably, by a value up to 7.8% when % of the manufacturing inventory holding cost is reduced to 50%. As compared to this, the retailer's inventory holding cost shows a minimum effect of 0.3% only for the total system's cost reduction when the retailer's inventory is reduced up to 50%.

Figure 3g,h portray the linear and fixed backorder costs, respectively. Both costs show a similar linear trend, and the overall system's cost is reduced by 2.7 and 2%, respectively, when the backorders are reduced up to 50%. Figure 3i illustrates the distributor's lot size effect on the system cost. Both show a similar linear trend and a reduction in the system cost by about 2.84% when the lot size is increased by 50%. As depicted in Figure 3j, there is an interesting relationship between the system cost and several shipments. Increasing the value of a few shipments by 50% increases the total cost of the system by 1.15%, while its value is decreased by 1.77% of its original value when the shipments are decreased by 50%. As shown in Figure 3k, there is a direct nonlinear relationship between the shipment lot size and the total cost of the overall system. By increasing or decreasing the shipment lot size, the total cost of the system is changed variably. Increasing the lot size value by 50% results in a 5% increase in the system cost, while it is reduced by 7.86% when the lot size value is reduced to 50%.

### 4.2. Numerical Example for Case 2

A manufacturer produces 550 items per year. The yearly demand placed by customers is 300 units. From the previous year's record, it is known that the manufacturer inventory carrying cost per product per unit of time is USD 50, and the retailer inventory carrying cost per product per unit of time is USD 5. The distributor inventory carrying cost per product per unit of time is USD 5, the linear backorder cost per product per unit of time is USD 10, the fixed backorder cost per product is USD 1, the manufacturing cost of a product is USD 7, the manufacturer production setup cost is USD 50, the retailer ordering cost is USD 5, and the backorder size is 33 items if the values of other parameters are assumed as a = 0.03 and b = 0.07 [4]. Table 5 shows the structures for item transport costs.

**Table 5.** Items unit transportation costs structure.

| Series of Items | Transport Costs in Dollar's/Units |
| --- | --- |
| $0.0 \leq L < 5.0$ | 4.0 |
| $5.0 \leq L < 10.0$ | 3.50 |
| $10.0 \leq L < 15.0$ | 3.20 |
| $15.0 \leq L$ | 3.0 |

The optimum number of shipments, optimum quantity, and total optimum cost can be calculated as follows:

$$A = 1 - \frac{a+b}{2} = 0.95$$

$$E_d = 1 - \frac{a+b}{2} - \frac{D_r}{P_r} = 0.4045$$

Step 1    $L_T^* = 9.61$ and $N^* = 6$
Step 2    $L_T^* < 15$ so go to step 3
Step 3

$$N_{up} = 6 \ and \ N_{lw} = \max\left\{1, L = \frac{\sqrt{\left[8E_d{}^2 P_r D_r C_m \left(K_m + B_f b_s\right)(P_r - D_r) + 4E_d P_r B_L b_s{}^2 AC_m (P_r - D_r)\right]}}{2E_d C_m (P_r - D_r)N}\right\}$$

$$N_{lw} = 3.380$$

Step 4    we solve the following for N = 4, 5, and 6
(a)    L = 12.640, N = 4.0, l = 2.0
(b)    *TC* (12.640, 4) = 4141.817 *TC* (16, 5) = 4147.717
(a)    N = 5, L= 10.870 l = 2
(b)    *TC* (15, 5) = 4129.46, *TC* (10.87, 5) = 4119.135
(a)    L= 9.610, N= 6, l = 1
(b)    *TC* (10, 6) = 4116.96 *TC* (9.61, 6) = 4115.632; *TC* (15, 6) = 4129.46
Step 5    the optimum solution is: *N* = 6, *L* = 10, and the total cost *TC* = $ 4116.960.

The results of the sensitivity analyses of case 2 for essential parameters are presented in Table 6 and explained in Figure 4a–i. Except for the production rate shown in Figure 4a, the number of shipments shown in Figure 4g, and lot size shown in Figure 4h, all other parameters show a linear relationship between the % change of a particular parameter and total system cost. The analysis of parameters shows a similar behavior and pattern as discussed in case 1. This shows the consistency of the obtained results in both cases.

**Table 6.** Sensitivity analysis for basic parameters of the model presented in case 2.

| Parameters | Variation | Variation (%) | Totals Costs | Variation in Total Cost (%) |
|---|---|---|---|---|
| $P_r$ | 687.5 | 25 | 4221.17 | 2.54 |
| | 825 | 50 | 4287.94 | 4.15 |
| | 275 | −50 | 3467.63 | −15.76 |
| | 412.5 | −25 | 3927.77 | −4.58 |
| $K_m$ | 62.5 | 25 | 4178.97 | 1.51 |
| | 75 | 50 | 4239.11 | 2.97 |
| | 25 | −50 | 3985.21 | −3.19 |
| | 37.5 | −25 | 4052.09 | −1.56 |
| $K_r$ | 6.25 | 25 | 4154.27 | 0.91 |
| | 7.5 | 50 | 4191.21 | 1.81 |
| | 2.5 | −50 | 4038.92 | −1.88 |
| | 3.75 | −25 | 4078.17 | −0.93 |
| $K_d$ | 6.25 | 25 | 4154.27 | 0.91 |
| | 7.5 | 50 | 4191.21 | 1.81 |
| | 2.5 | −50 | 4038.92 | −1.88 |
| | 3.75 | −25 | 4078.17 | −0.93 |
| $C_r$ | 6.25 | 25 | 4123.01 | 0.15 |
| | 7.5 | 50 | 4129.44 | 0.31 |
| | 2.5 | −50 | 4103.84 | −0.31 |
| | 3.75 | −25 | 4110.21 | −0.15 |

**Table 6.** *Cont.*

| Parameters | Variation | Variation (%) | Totals Costs | Variation in Total Cost (%) |
|---|---|---|---|---|
| $C_d$ | 6.25 | 25 | 4116.76 | 0.003 |
| | 7.5 | 50 | 4116.94 | 0.008 |
| | 2.5 | −50 | 4116.34 | −0.006 |
| | 3.75 | −25 | 4116.46 | −0.003 |
| $B_L$ | 12.5 | 25 | 4157.99 | 1.005 |
| | 15 | 50 | 4198.21 | 1.98 |
| | 5 | −50 | 4030.96 | −2.08 |
| | 7.5 | −25 | 4074.28 | −1.02 |
| $M_c$ | 8.75 | 25 | 4739.92 | 13.39 |
| | 10.5 | 50 | 5291.11 | 26.78 |
| | 3.5 | −50 | 3086.16 | −26.78 |
| | 5.25 | −25 | 3637.40 | −13.39 |
| a | 0.0375 | 25 | 4123.62 | 0.17 |
| | 0.045 | 50 | 4130.64 | 0.34 |
| | 0.015 | −50 | 4102.56 | −0.34 |
| | 0.0225 | −25 | 4109.58 | −0.17 |
| b | 0.0875 | 25 | 4132.99 | 0.39 |
| | 0.105 | 50 | 4149.37 | 0.79 |
| | 0.045 | −50 | 4083.83 | −0.79 |
| | 0.0525 | −25 | 4100.22 | −0.39 |
| $D_r$ | 375 | 25 | 4688.58 | 13.89 |
| | 450 | 50 | 5209.37 | 26.54 |
| | 150 | −50 | 2805.03 | −31.86 |
| | 225 | −25 | 3491.02 | −15.19 |
| N | 7.5 | 25 | 4134.28 | 0.42 |
| | 9 | 50 | 4163.73 | 1.14 |
| | 3 | −50 | 4189.16 | 1.76 |
| | 4.5 | −25 | 4122.92 | 0.15 |
| L | 12.5 | 25 | 4169.18 | 1.28 |
| | 15 | 50 | 4284.07 | 4.07 |
| | 5 | −50 | 4567.03 | 10.95 |
| | 7.5 | −25 | 4186.92 | 1.71 |
| $T_u$ | 4.375 | 25 | 4201.40 | 0.02 |
| | 5.25 | 50 | 4214.16 | 0.04 |
| | 1.75 | −50 | 4163.16 | −0.04 |
| | 2.625 | −25 | 4175.92 | −0.02 |

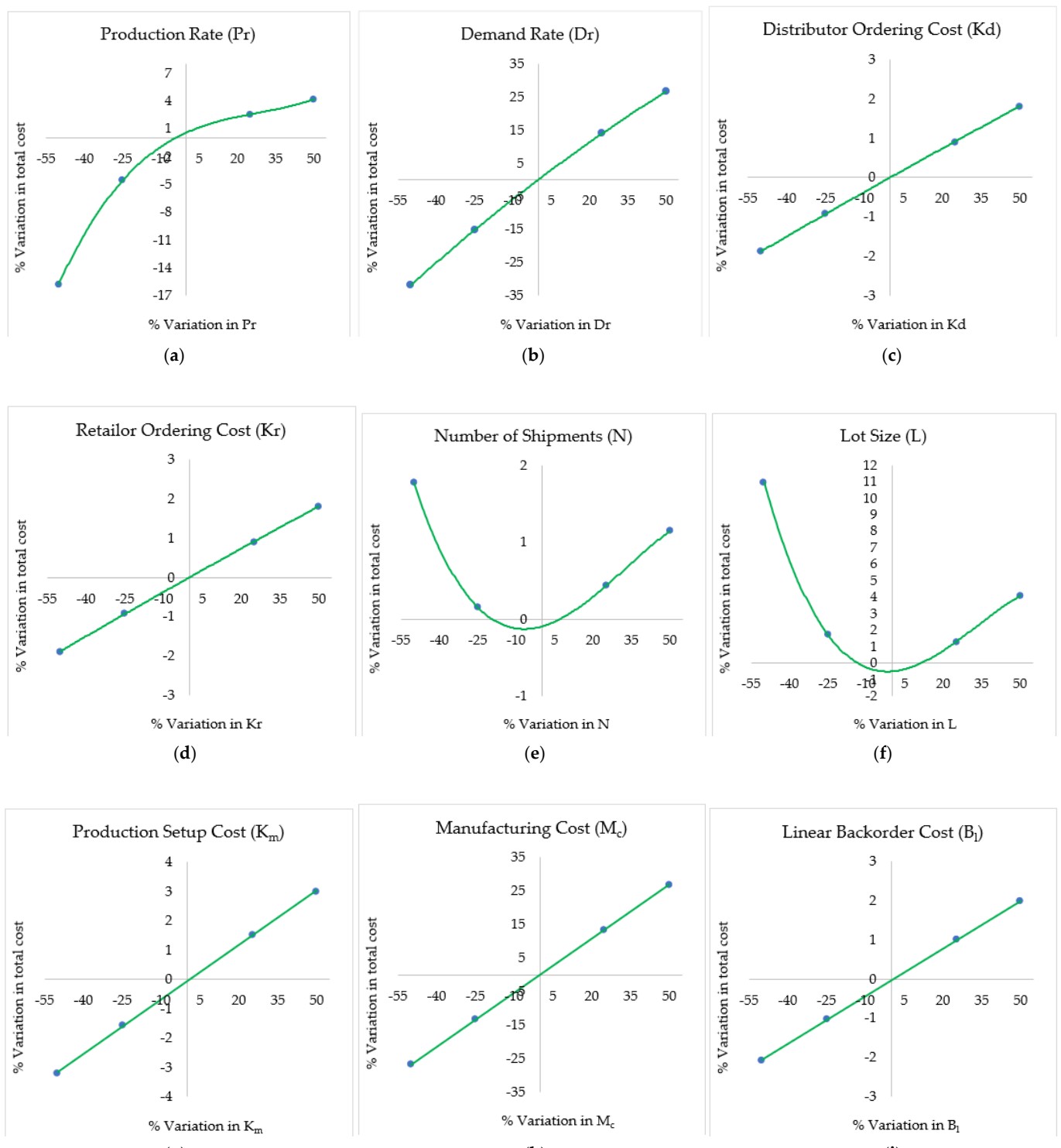

**Figure 4.** (**a–i** from top left to bottom right). Sensitivity analyses of important parameters (Case 2).

## 5. Conclusions and Future Recommendations

The model developed for the optimal shipment of products and numerical findings contributes to the knowledge in the fields of production and supply chain management. The main contributions of the presented research are as follows:

In this research, a shipment size model for the manufacturer, distributor, and retailer with an equal-sized shipping policy is developed for the imperfect production system. The all-unit reduction transportation cost structure has been evaluated for the proposed

model. The optimum solution methods are also developed and analyzed. From the solution procedure, it can be observed that the shipment decision varies as the transportation cost is incorporated into the system. The model presented in Case-II resulted in a reduced total system cost by 30.70, which means the total cost has been reduced by more than 1.00%. This shows the effectiveness of the proposed development for an imperfect production system. Similarly, in the model presented in Case-II, the total system cost has been reduced to 76.29, which means that the total cost has been reduced by more than 2.00%. This also highlights an advantage of the developed model. These reductions in total costs are just based on the numerical values assumed in the presented examples. The assumed numeric values are just for model validation purposes and in a real scenario are much lower (demand: just 300 units per year, production 550 per year, and so on) than the real cases values. These results can only be applied and compared with the real cases if the numeric values in the computations are taken as per actual data.

The numerical computations and sensitivity analysis are performed to point out the specifications of this work. From sensitivity analysis, it is found that increasing the values of certain parameters, like the fixed setup cost (K), ordering cost (Km, Kr, and Kd), unit inventory carrying cost (Cm, Cr, and Cd), fixed cost per backorder (Bf), linear backorder cost (BL), unit manufacturing cost (Mc), and demand (Dr) results in increased values of the total cost of the system. By decreasing the values of these parameters, the value of the total cost of the system is decreased as well. Different results ($-50\%$ to $+50\%$) can be estimated for (L) and (N) by changing the shipment lot size (L) and the number of shipments (N). Furthermore, it is found that demand (Dr) and unit manufacturing cost (Mc) are the most sensitive parameters compared to all other parameters.

In this research, a single type of production system is considered to produce only one type of item. In actual production systems, several products can be produced simultaneously in multistage production systems. So, the proposed model can be extended considering multiple products and a multistage production system. In this model, we have considered one retailer, manufacturer, and distributor, while many retailers, manufacturers, and distributors can be considered to extend the scope of this model. In this shipment policy, we have considered a single manufacturer, single retailer, and single distributor. Future research will consider a situation where multiple suppliers may exist along with the manufacturer, retailer, and distributor. In addition, the research can be extended by considering the variable demand rate.

**Author Contributions:** Conceptualization, W.S., M.U. and M.H.; methodology, W.S. and M.H.; validation, R.K.; formal analysis, W.S. and M.H.; investigation, W.S. and R.K.; writing-original draft preparation, W.S. and M.H.; writing-review and editing, M.U. and R.K.; supervision, M.U. All authors have read and agreed to the published version of the manuscript.

**Funding:** This research received no external funding.

**Institutional Review Board Statement:** Not applicable.

**Informed Consent Statement:** Not applicable.

**Data Availability Statement:** Data will be available on request.

**Conflicts of Interest:** The authors declare no conflict of interest.

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
