# Peer review of "Developing a Comprehensive Shipment Policy through Modified EPQ Model Considering Process Imperfections, Transportation Cost, and Backorders"

_logistics_

Round 1
Reviewer 1 Report
I have completely reviewed the article. I consider that the author’s proposed model is superior to the related existing works. The setting of the mathematical model is excellent. However, the writing skill needs to be improved. Some descriptions for explaining the proposed model seem not enough. The presentation of mathematical inferences is also inappropriate. I think the study can be accepted in this journal if the author can improve the issues as follows:
- Figure 2 illustrated the proposed EOQ model in detail. The author tried to use a simple way to present the logical process of the calculation. However, it seems unnecessary to present the mathematical formulas regarding the related costs. It is suggested to reserve the logical process of the calculation and remove these mathematical formulas. The formulas of the related costs can be presented in Section 3 and explained them in detail. I consider that it can increase the readability of the study.
- The author needs to provide more descriptions for explaining every part of the proposed model at the beginning of Section 3. In other words, the author needs to tell the story of the study to attract the readers. Accordingly, the readers can understand the reason why the proposed model can provide a useful solution for the critical issue in practice.
- Please check the proof of optimizing TC(L,N). We believe that your model can show the convexity to the two decision variables. However, obviously, the description of the Hessian matrix may be incorrect.
- Some of the mathematical inference steps need to be omitted. For example, Equations (31), (32), (33), (34), (35) and (36) are not necessary for the professional readers. Similarly, Equations (40), (41) and (42) can be omitted. The author should give clear mathematical presentations with adequate explanations to readers.
- It is unnecessary to present the whole calculation process with numbers on pages 14 and 15. We understand that the author wants to give full comprehension to readers. However, it seems unnecessary to professional readers. I suggested that the author may remove these numbers and keep the mathematical formulas in a concise way.
Reviewer 2 Report
The authors present the design of a new shipment policy considering the imperfections in the production processes, transportation cost, production inventory, defective items, and backorders. The subject is interesting, but there are many major concerns as follows:
- Introduction
Please cite some studies that have expanded the EOQ and EOP model (Line 37-38).
The authors mention that (line 38-39) “The research studies have shown that small perturbations in parameters of EOQ and EPQ models do not impose any significant impact on the solution of a problem”. Are the authors referring to the Harris and Taft research studies, or to other studies that have analyzed the stability of the EOQ and EOP? If so, please mention those studies.
Please, divide the paragraph (line 43-71) into two or more paragraphs, to make it easier to read.
The introduction section should end by presenting the objective of the article, the main contributions and the reasoning why shipment with imperfection in the process considering distributors should be considered, which is the differentiating factor with respect to studies in the literature according to Table 1. Likewise, present the content of the following sections of the article.
- Model Specifications
It is recommended to present the model parameters before Figure 2 in order to understand the cost formulations presented there.
The sentence (line 164-166) “The suggested work expands the most recent study of Hayat et al. [30] by incorporating a shipping policy in the defective manufacturing system.” It should appear right after Table 1, and not in section 2.
The authors must justify and cite the documents from which they obtained the cost formulations presented in Figure 2, and declare which formulations are original in this article and which are used by other authors. This is very important to highlight the originality of the study.
- Model Formulation
In this section, the different formulas presented (From Equation 1 to Equation 44) must be cited or mentioned, and the meaning or steps represented in said formulas must be explained. For example, explain why in Equation 2 Qs is replaced by NL.
Check in Equation 2 the expression (.2−?1). Should it be (2−?)?
I consider that this section should be rewritten following the principles of scientific article writing, thus facilitating the understanding of the model and the meaning of the presented equations. For example, authors should not write “To simplify the above equation” (Line 256). It should be mentioned which specific equation the authors are referring to. Are the authors referring to Equation 16?
What does Equation 44 mean? Authors should mention what the result of that equation is for.
It is not clear how the sensitivity analysis was performed. What were the values of the parameters that were used as a basis to build Figure 3? For example, what values does Dr, Km, Kd, Mc, Cm, Bl, Bf, N, L, Ld take while the values of the parameter Pr vary?
The authors should improve the layout of Figure 3 so that all graphs appear on four lines instead of five lines.
Please mention the software used to calculate the model and to generate the sensitivity analyses.
- Numerical Computation and Sensitivity Analysis
Please improve the number format on Table 4 and Table 6 by reducing the number of decimal places for the “Total costs” and “Variation in total cost (%)” columns.
In the parameters, it is necessary to describe what A means, and in turn to explain what the parameters a and b that make up A mean. The authors assume that a = 0.03 and b = 0.07. What are they based on to make this assumption? Why don't authors do a sensitivity analysis of the model considering different values of A, a and b?
Is a 1% or 2% reduction in total costs considered significant? Please mention similar studies that yield similar or lower figures to validate the significance of the costs savings.
Do the values of the numerical examples correspond to any real case? Explain what the authors used to assign the values to the parameters in Example 1 and 2.
Round 2
Reviewer 2 Report
The authors have accepted the suggestions and have presented the necessary corrections to obtain a quality article.